# Exploring Determinants of Interdisciplinary Collaboration within a Geriatric Oncology Setting: A Mixed-Method Study

**DOI:** 10.3390/cancers14061386

**Published:** 2022-03-09

**Authors:** Marion Barrault-Couchouron, Noemi Micheli, Pierre Soubeyran

**Affiliations:** 1Team PHARES, Centre INSERM, U1219, Université de Bordeaux, 146 rue Leo Saignat, CEDEX, 33076 Bordeaux, France; 2LabPsy, EA4139, Université de Bordeaux, 3ter Place de la Victoire, CEDEX, 33076 Bordeaux, France; n.micheli@bordeaux.unicancer.fr; 3Institut Bergonié, 229 Cours de l’Argonne, CEDEX, 33076 Bordeaux, France; p.soubeyran@bordeaux.unicancer.fr

**Keywords:** geriatric oncology, oncogeriatric care, elderly cancer patients, physician perception, geriatrician perception, mix-method study, interdisciplinary collaboration

## Abstract

**Simple Summary:**

Collaboration between oncologists and geriatricians has been shown to improve the quality of elderly cancer patient care. However, previous research has revealed how interpersonal factors might hinder this interdisciplinary work. This study aims to assess sprocessual and contextual determinants of the collaboration between these two disciplines, including shared time and routines, medical decision criteria and perceptions of age and needs of elderly patients. These aspects are important to develop a more efficient patient-centered approach in oncogeriatric care and improve collaboration between the different disciplines involved.

**Abstract:**

Therapeutic challenges regarding the population of elderly cancer patients and their heterogeneity lead to the need to implement person-centered approaches in order to optimize care strategies and adapt oncology treatments to each pattern of aging. The International Society of Geriatric Oncology recommends a multidisciplinary evaluation of these patients and the use of screening tools prior to the initiation of treatments. However, previous research shows a poor implementation of these recommendations in geriatric oncology. Although some studies have identified how different perceptions of geriatric oncology might hinder routine teamwork, little is known about the impact of other factors on promoting the collaboration between the two specialties. This mixed-method exploratory study used an online questionnaire to assess the perception of a group of 22 geriatricians and oncology physicians on different determinants of oncology care and teamwork. In this sample, older oncology patients benefited from geriatric care. However, there was a variability regarding age criteria and a limited use of screening tools. The multidimensional framework for interprofessional teamwork by Reeves has been used to analyze some of the determinants of the collaboration between oncology physicians and geriatricians. This study has identified systematic issues to consider when promoting communication and common values between the two disciplines, including available resources in terms of shared time, space and routine actions.

## 1. Introduction

Cancer is well recognized as a disease of old age, with approximately 60% of all cancers occurring in people older than 65 years of age [1]. The older cancer population is increasing worldwide and its heterogeneity creates new therapeutic challenges related to the limited medical knowledge of the disease at this age and to the need for adapted treatment in older patients [1,2,3]. It is estimated that by 2030, 70% of cancers and 85% of cancer-related deaths will occur in the elderly who are over 65 years of age [1]. Research has highlighted various problems related to older cancer patients’ care, such as late diagnosis and incomplete investigations [3,4], less intensive treatment than in younger populations [5] and the lack of participation of elderly cancer patients in clinical trials [6]. Thus, the fact that elderly cancer patients are less likely to receive optimal or standard care is a major concern, constituting an important focus for policy makers [7]. Although a contribution of age-associated conditions, such as geriatric syndromes or comorbidities, has to be taken into consideration, the variation of treatments compared to younger counterparts is not always fully explained by these prognostic factors, and could be the result of an inappropriate emphasis on chronological age, resulting in potentially denied curative treatments [8]. This aspect is particularly problematic, as research has demonstrated that chronological age is an imperfect marker for the presence of impairment and frailty in older cancer patients [9]. 

Modern management in elderly cancer patients “demands patient-centered treatment, assessing frailty rather than chronological age” [10]. Therefore, in the last two decades, an emphasis has been placed on the creation of geriatric oncology programs and on a collaborative approach within different specialties to optimize therapeutic strategies and to adapt treatment decisions [11,12,13,14]. One example of these programs is the Pilot Oncogeriatric Coordination Units (UPCOGs) created by the French National Cancer Institute (INCA). In these units, rather than creating a new medical specialty, the optimization of elderly cancer patients’ care is based on the creation of a multidisciplinary team and on a collaborative partnership between cancer specialists and geriatricians. 

To evaluate vulnerabilities that are not routinely captured in oncology assessments [15], the International Society of Geriatric Oncology and the ASCO Guideline for Geriatric Oncology currently recommend the use of geriatric assessment (GA) prior to initiation of oncologic treatment [16,17]. GA is a validated multidimensional tool to assess the overall health status of an older adult in cancer care and includes the assessment of a range of domains affecting this population, such as evaluation of physical function, comorbidity, polypharmacy, cognition, psychological status, social support and nutritional status [9,18]. Research has shown that GA not only guides the clinical decision-making process [19,20], it can also predict chemotherapy-related toxicity [20,21], improve patient satisfaction [22] and positively affect treatment completion [20]. As a full GA can be time-consuming, the G8 screening tool has been developed to identify patients with cancer over 70 years of age requiring a full GA and multidisciplinary approach [23,24]. The G8 tool is easy and quick to administer (median time of 5 min) and consists of eight items covering multiple GA domains [25]. Its sensitivity is 65–92% and has been shown to be more than 80% in many studies detecting impairments via a full GA [26]. Two systematic reviews concluded that the G8 was one of the most robust screening tools currently available [27,28]. However, current evidence of the use of GA tools with older patients remains minimal [9,18,29,30].

So far, few studies have explored how oncology physicians and geriatricians perceive their collaboration e.g., [10,11,31]. For example, Sifer-Rivière et al. [32] reported that both oncology specialists and geriatricians have heterogeneous and ambiguous expectations about their collaboration in clinical routine. The authors explained the inadequate dialogue between these specialties by a lack of common professional values, offering a better understanding of the barriers to an effective oncogeriatric collaboration. However, previous studies have only focused on interpersonal dynamics, ignoring other important factors [33]. 

Interprofessional collaboration is a term referring to a broad activity whereby different health and social care professions regularly come together to improve the delivery of patient care [33] (p. 4). In their multidimensional conceptual framework for interprofessional teamwork, Reeves et al. [33] identified several factors impacting collaborations, dividing them into four categories: relational, processual, organizational and contextual. Although the collaboration between geriatricians and oncology physicians could be better defined with the term “interdisciplinary”, these two disciplines vary significantly in their goals. “Oncology physicians focus on assessment of cancer variables, such as tumor biology and stage, and develop cancer-specific treatment plans. On the other hand, geriatricians assess physiologic age and functional status, and focus on optimizing an individual’s independence” [34]. Therefore, the present study explores physicians’ perceptions on SIOG recommended activities and on interdisciplinary collaboration, looking at some of the organizational, processual and contextual determinants of the conceptual framework for interprofessional teamwork by Reeves et al. (Figure 1) [33]. This will provide a deeper understanding of the different dynamics hindering or promoting coordinated oncogeriatric medical practice as well as a better understanding of the perceived challenges regarding the implementation of routine geriatric-based competencies within the oncology setting and the care of elderly cancer patients. The implementation of a formalized working structure not only offers concrete guidelines for oncogeriatric practice, it also promotes medical objective treatment decisions. 

## 2. Materials and Methods

For this exploratory mixed-method study, an online questionnaire was addressed to physicians working in oncology and geriatrics. 

In light of the specific multidimensional nature of the study object, a triangulated methodology was implemented in order to enrich the data with both a qualitative and quantitative point of view [35,36]. In fact, the use of dual perspectives is a strategy that can offset bias [37] and provides complementary data suitable for exploratory studies [38]. The aim was to depict a more comprehensive view on contextual, processual and organizational determinants of the collaboration between oncology physicians and geriatricians by combining different types of data. 

The qualitative part of this study was realized with the elaboration of open-ended questions assessing physicians’ perceptions on the needs of elderly cancer patients, on the collaboration between oncologists and geriatricians and on the obstacles to such a collaboration. 

The quantitative part of this study was implemented through the administration of closed-ended questions, which were used to assess medical decision criteria and treatment preferences, the participation in interdisciplinary meetings and physicians’ perceptions of elderly cancer patients.

The research team consisted of two women and one man, all experienced in oncology inpatient care. At the time of the study, the researchers responsible for the qualitative data coding were working in a comprehensive cancer center, and one held a PhD. 

As part of the ONCODAGE project, this ancillary study was approved by the Council for the Protection of Persons. Data collection were carried out in 2012. 

### 2.1. Population 

Participants included in this research were selected using purposive sampling and were physicians working with cancer patients in the oncology and geriatric departments or physicians working in the Pilot Onco-Geriatric Coordination Units (UPCOG) participating in the French ONCODAGE Prospective Multicenter Cohort Study. The main objective of the ONCODAGE study was the validation of the G8 assessment tool. Of the 41 physicians invited to participate in this study, 22 completed the questionnaire.

### 2.2. Procedure

All physicians collaborating in the ONCODAGE cohort were contacted by an email explaining the objectives of the present study and inviting them to participate. Physicians were informed that the purpose of the study was to explore the collaboration between oncology physicians and geriatricians and their perceptions of elderly cancer patients’ care. Following informed consent, a hypertext link redirected them to the online questionnaire. 

### 2.3. Questionnaire

The questionnaire included a series of open and closed questions.

Closed-ended questions assessed different domains, including:Sociodemographic data: age, sex, profession, additional degree in geriatrics or oncology, workplace and years of professional experience.Relationship and perception of elderly cancer patients: this category of questions assessed physicians’ perceptions about the beginning of elderhood (e.g., “From what age do you consider a patient to be old?”), the number of elderly patients for each physician in the last month (“In the past month, how many elderly patients have you cared for?”), and the proportion of patients shared with the other specialty (“What is the proportion of your elderly patients currently under the care of a geriatrician?” or “What is the proportion of your elderly patients currently under the care of an oncologist?”).Medical decision criteria and treatment preference (e.g., “How do you make your medical decisions?”(in multidisciplinary meeting, after consulting other specialists, alone or other); “What are the factors that you take into consideration in your decision-making process for treatment choices?”(medical history, characteristics of the tumor, patient’s desire, age, psychosocial factors or impossibility of using treatment due to patient fragility); “What type of treatment do you usually prefer in the care of elderly cancer patients?” (surgery, hormone therapy, radiotherapy, chemotherapy or other).Presence and participation in interdisciplinary meetings (IDMs): two questions assessed the frequency of scheduled IDMs (e.g., “In your department, how often are interdisciplinary meetings with all doctors involved in the care of elderly patients and other professionals providing support for these patients held?” (weekly, bimonthly, monthly, less than once a month, there are no multidisciplinary meetings, do not know or no answer) and the frequency of participation for each physician (“How often do you participate?” (systematically, frequently (2 times out of 3), rarely (1 time out of 3), never, do not know or no answer).

Open ended questions concerned:Physicians’ perceptions of older cancer patients’ needs (e.g., “What do you think are the specific needs of elderly patients treated for cancer?”);Use of GA tools (“Do you use geriatric assessment in your practice?”, “What kind of GA do you use?”);The perception of the interdisciplinary collaboration between oncology physicians and geriatricians (“What does working in collaboration with doctors working in oncology (oncologists, surgeons, anaesthetists and radiotherapists) bring to you?”, “What does working in collaboration with the geriatrician bring to you?”);The main obstacles of the collaboration between oncologists and geriatricians (e.g., “What, according to you, are the main obstacles to a full collaboration between oncologists and geriatricians or between your service and oncogeriatric care services?”).

### 2.4. Data Analysis 

Quantitative data analysis, including frequencies, means and standard deviations, was analyzed using SPSS 19. Qualitative data were analyzed using framework analysis [36]. This approach was designed for applied policy research allowing the identification of a priori and emergent themes. The qualitative variables were analyzed using NVivo-9 software (QSR International, Boston, MA, USA) according to the methodological recommendations for this type of mixed protocol [39]. The complete transcript of the qualitative data was uploaded to the NVivo-9 qualitative analysis software, thus facilitating the analysis of common themes. In order to ensure the validity and reliability of the analysis, a double coding of 15% of the coding material was carried out, first by the same coder and then by a second coder. This procedure aimed to achieve high rates of intra-judge and inter-judge loyalty. Data description and integration were accomplished through thematic content analysis using the multidimensional conceptual framework of interprofessional teamwork by Reeves et al. [33].

## 3. Results

Results are organized into three parts. The first is a description of the study population, the second describes important factors regarding physicians’ care of elderly cancer patients and the third examines oncogeriatric medical practice and different determinants of the collaboration between oncology physicians and geriatricians. 

### 3.1. Sample Characteristics

Of the 22 participants, 7 geriatricians worked in the geriatric setting and 15 physicians worked in the oncology setting (9 oncologists, 2 radiotherapists, 2 surgeons, 1 gastroenterologist and 1 anesthetist). Overall, physicians had a mean age of 43.8 years (range = 33–61, SD = 8.2) and had a mean of 14.7 years (range = 3–36, SD = 7.9) in professional experience. Table 1 lists their main characteristics, including age, years of professional experience and additional university degrees in oncology or geriatrics. 

#### 3.1.1. Geriatric Service

The median age of participants working in the geriatrics setting was 45.5 years (range = 34–60, SD = 9.6). The median of their working experience was 11.7 years (range = 3–15 years, SD = 4.3). Of the seven geriatricians, four reported an additional university degree in oncology. 

#### 3.1.2. Oncology Service 

Median age of oncology physicians was 43.79 years (range = 33–61, SD = 8.2), median working experience was 16.3 years (range = 6–36 years, SD = 8.9), and, out of 15, five oncology physicians reported an additional university degree in geriatrics. 

### 3.2. Physicians’ Care of Elderly Cancer Patients

#### 3.2.1. Oncogeriatric Care

Overall, oncology physicians reported limited oncogeriatric care for their patients, with only 38% of their elderly cancer patients receiving geriatric care. Geriatricians on the other hand, reported greater oncogeriatric care for their older patients, with 67% of their patients receiving oncogeriatric care (Table 2).

#### 3.2.2. Perceptions of Elderly Age 

Both geriatricians and oncology specialists reported variability in the beginning of the age criterion for which they perceived their patients as being elderly. Responses varied from 60 years to 85 years. There were differences in the representations for the beginning of elderhood between oncologists and geriatrics. The average age for oncology physicians was 72 years old (min 60, max 80, SD 4.55), while the average age for geriatrics was 75.2 (min 70, max 85, SD 6.10). Therefore, the minimum age at which geriatricians perceived a patient to be old was 10 years older than the oncology physicians’ perceptions. 

#### 3.2.3. Medical Decision Criteria and Treatment Preference 

The most preferred treatment options reported by both oncology physicians and geriatricians were (from most preferred to least preferred): surgery, hormonotherapy, radiotherapy and chemotherapy.

The majority of oncology physicians reported tumor characteristics and medical history as the most important criteria to determine therapeutic proposals, followed by a patient’s desire and age. Geriatricians, on the other hand, identified a patient’s desire and medical history as the most important criteria, followed by tumor characteristics and psychosocial factors. If oncology physicians frequently identified “age” as one of the decisive criteria for therapeutic proposals, this criterion was not reported by geriatricians. 

#### 3.2.4. Perceptions of the Specific Needs of Elderly Cancer Patients

All geriatricians shared the need for a patient-centered approach in order to “*maintain patient’s quality of life*” and “*adapt patient’s need for care to his/her level of autonomy*”. In terms of the clinical decision-making process, they described the need for “*using a global medical and psychosocial assessment*”, taking into account a patient’s desire. In addition, geriatricians identified the elderly patient’s need to be understood and to feel confident. 

Oncology physicians agreed on the need for multidisciplinary elderly patients’ care. Moreover, oncology physicians identified care-centered needs, specifically: “*cooperation with patient’s family and patient’s social environments*” and “*coordination with other health care professionals to secure treatment and after-care of the older patient*”. In addition, “*increased monitoring to prevent potential side effects*” was also mentioned multiple times as one of the care priorities regarding older patients’ fragility. 

### 3.3. Oncogeriatric Medical Practice and Interdisciplinarity

#### 3.3.1. Use of Geriatric Screening Tools in the Oncology Setting 

Overall, oncology physicians reported limited use of screening tools, with only one third integrating GA in their routine practice. Only oncology physicians with supplementary education in geriatrics reported the use of GA prior to the decision-making process. Of the five oncology physicians using GA, three reported using the G8 tool (Table 3). The main reason for oncology physicians not using GA was the collaboration with geriatricians: “*GA is performed by the geriatric team*”; “*I have a geriatrician who sees my patients*”; “*I show my patient cases aged 70 years or older to our geriatricians*”. In total, seven oncology physicians reported a habit of using GA, including the use of screening tools (e.g., VES-13, G8), multi-GA (e.g., ADL, IADL, MNA and “get up and go”) and clinical geriatric evaluation. 

#### 3.3.2. Interdisciplinary Meetings (IDMs)

All physicians reported making their medical decisions about elderly patients after consulting other specialties (e.g., direct interdisciplinary collaboration and interdisciplinary meetings). However, both geriatricians and oncology physicians reported a variability regarding the frequency in which interdisciplinary meetings were organized. Overall, 33% of participants reported not having IDMs. 

In the group of oncology physicians (*n* = 15), 7% stated having IDMs once a month and 60% had IDMs weekly. Moreover, five physicians out of nine attended IDMs systematically, two frequently and two rarely (Table 3). 

In the group of geriatricians (*n* = 7), 17% systematically participated in bimonthly IDMs and 50% (*n* = 3) had weekly IDMs (to which one attended rarely, one frequently and one systematically) (Table 3). 

The main barrier to routine attendance at IDMs was described in terms of availability. In addition, both geriatricians and oncology physicians reported a “*lack of older patients in their active file*”, “*lack of structured IDMs*”, and “*difficulties in interdisciplinary communication*” as important elements hindering their participation in IDMs. 

#### 3.3.3. Perceptions on Oncogeriatric Medical Practice 

Overall, geriatricians described the importance of a “*multidisciplinary approach*” in order to provide therapeutic proposals and tailored global care plans. Only one geriatrician described the specific contribution of oncologic competencies within geriatrics: “*Increased knowledge of clinical oncology provides a better understanding of medical decisions, treatment choices and possible side effects. Sharing medical knowledge in oncology and geriatrics promotes the overall care of the older patient with cancer and helps preserving the patient’s quality of life. In addition, we are able to better inform the patient about our medical choices and offer him/her a better understanding of his/her illness and our therapeutic proposals*”.

Some oncology physicians mentioned no current benefits due to a lack of a structured working relationship with geriatricians. These elements were also mentioned in terms of main barriers to effective collaboration by all oncology physicians: “*absence of geriatricians during IDMs*”, “*the lack of geriatricians in oncology setting*” as well as their “*lack of availability in time*”, and an overall lack of “*coordination of the structure’s organization*”. Some physicians also reported the “*lack of common language or culture*” and the “*non-recognition of the contribution of geriatric-based competencies within the oncology setting*”. 

Regarding benefits of oncogeriatric medical practice, physicians described the contribution of “*a different and more global vision in the management of the older patient*” and the patient’s specific needs, particularly in the more complex cases. In addition, sharing knowledge and competencies was described to facilitate common research projects.

## 4. Discussion

Since 2005 a two-step approach for using GA has been recommended by the International Society of Geriatric Oncology (SIOG) in order to promote a collaborative person-centered approach in the care of older patients with cancer [40,41]. This approach includes a short screening test of every older patient presenting for treatment, and a multidisciplinary evaluation in the presence of a geriatrician for patients screening at risk. This should translate into the adoption of an identical line of conduct by cancer specialists regarding the use of geriatric screening tools, instead of relying solely on the physician’s clinical impression. However, research has demonstrated a high diversity in treatment approaches among oncology physicians [3,4,5]. In line with previous studies, in the current sample, the use of GA by oncology physicians was limited and only 38% of elderly cancer patients benefited from a geriatric intervention. Only oncology physicians with supplementary education in geriatrics reported the use of GA prior to the decision-making process.

In theory, the organization of structured oncogeriatric medical activity is based on age criterion. Present results are in line with previous research demonstrating variability in physicians’ perceptions regarding the age at which they perceive their patients as being elderly [31,38]. In fact, in this study’s sample, the minimum age at which geriatricians perceived a patient to be old was 10 years older than the perception of the group of oncology physicians. This aspect seems to be even more relevant considering the fact that for oncology physicians, age was an important criterion among medical factors influencing treatment decisions. The noted lack of formalized age criterion and limited application of GA in the decision-making process for older patients with cancer seems to support previous research, recognizing that physicians’ variable perceptions of the elderly affect medical treatment [26,42,43]. 

Taking a step forward from previous studies, which showed how interpersonal dynamics can have an impact on the collaboration between geriatricians and oncology physicians [31,32], the current study aimed to look at other determinants influencing this work relationship. Reeves et al. [33] identify routines, time and space as some of the processual factors that could affect interprofessional communication in teamwork. Spatial and time boundaries may promote profession-specific tasks and inhibit professional teamwork [40]. In addition, it is generally agreed that more shared time is needed to develop mutual understanding, trust and respect in teams. In this study’s sample, all physicians agreed on the need for multidisciplinary care of older patients, but the processual determinants for such collaboration were minimal. Limited shared time between geriatricians and oncology professionals was noted as their attendance at interdisciplinary meetings was variable and IDMs often conflicted with their profession-specific schedules. 

Reeves et al. [33] also explained the need for routine activities in guiding interprofessional communication by clarifying physicians’ expectations, in particular when processes in delivery of care are becoming more complex (e.g., aging population), and processes of task-shifting occurs (e.g., creation of new roles for geriatricians and oncologists). In the sample of this study, physicians described the lack of common language and the non-recognition of the contribution of geriatric-based competencies within the oncology setting as important barriers to successful teamwork. Diversity regarding profession-based values among geriatricians (i.e., global patient-centered approach) and oncology specialists (i.e., illness-centered approach) within a team is considered as crucial in optimizing tailored care for older patients with cancer. However, research shows that in reality, collaborating together from different perspectives can be difficult and challenging to achieve [7,31,44,45]. This aspect might be even more relevant considering that in the current sample, the priority of criteria between oncology physicians and geriatricians on the factors impacting medical decisions were different, with oncology physicians prioritizing tumor characteristics and medical history, and geriatricians prioritizing the patient’s desire. 

Within the contextual determinants for interprofessional teamwork, Reeves et al. [33] explain culture as: “the meanings and perceptions different team members attach to their team as well as their interprofessional interactions”(p. 73). Meaning is constructed between different professionals within teams and determines a shared agreement regarding interprofessional team practice. If the lack of common culture between geriatricians and oncology physicians currently challenges effective teamwork, this might be associated with a lack of shared activities. In fact, educational and professional experiences, including the implementation of routine GA procedures, could reinforce common values, approaches and languages for each profession [45,46]. 

This study carries several limitations. First, being an ancillary study of the ONCODAGE Prospective Multicenter Cohort Study, not all outcomes of interest could be explored. In fact, as suggested by the framework by Reeves et al. [33], other determinants should be assessed in order to fully comprehend the functioning of interprofessional teamwork. A degree of responder bias has to be taken into consideration in light of the main study objective being the validation of the G8 tool. A further aspect to consider is the small sample size of the study and the low response rate that has been registered, which could have affected the results. This study aimed to conduct an initial exploration of some important factors in the collaboration between oncologists and geriatricians and in the oncogeriatric care of elderly cancer patients, but even implied causality should be viewed cautiously. Definitive answers to the questions raised here will require further studies. Finally, in spite of the fact that the data reported in this study were collected in 2012, the findings are consistent with more recent literature. 

## 5. Conclusions

Based on the multidimensional framework used as a guide for presenting some of the results, several factors to consider when promoting interdisciplinary collaboration in oncogeriatric care were found. The lack of communication and common values between geriatricians and oncology physicians could be associated with some important processual determinants, such as the limited shared time and routines between the two disciplines. Health policy makers need to be aware of these factors and put in place clear actions to improve oncogeriatric collaboration. Structured training and incentives favoring the integration of geriatric principles into oncology training need to be encouraged. Joint activities during the training of doctors could be evaluated. Nevertheless, a common age criterion and the systematic use of the G8 screening tool might help to reduce the variability of treatment in elderly cancer patients and to promote a person-centered approach in their care. 

## Figures and Tables

**Figure 1 cancers-14-01386-f001:**
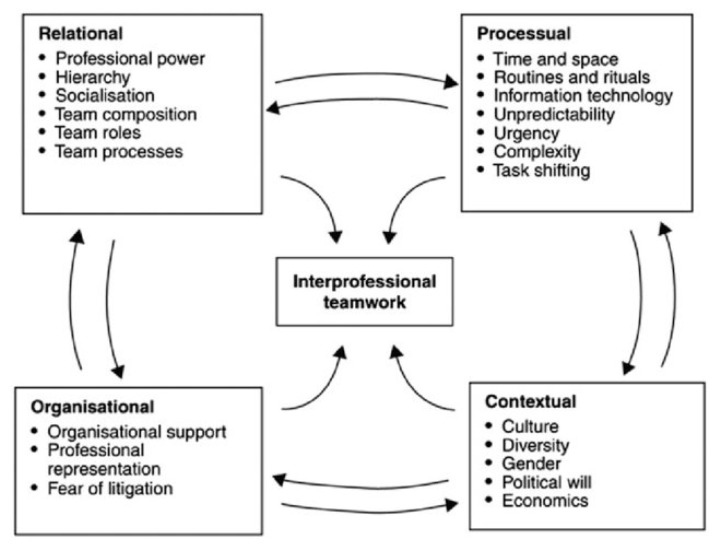
Conceptual Framework for Interprofessional Teamwork. Source: Reeves et al. [33], IGI Global, reprinted by permission.

**Table 1 cancers-14-01386-t001:** Sample characteristics.

Characteristic	Geriatricians (*n* = 7)	Oncology Physicians (*n* = 15)	Total (*n* = 22)
Age	45.5 {9.6}	43.7 {8.2}	43.8 {8.2}
Sex			
Female	(57%)	(53%)	-
Number of professionals with an additional degree:			
-in oncology	4	-	4
-in geriatrics	-	5	5
Years practicing:			
-oncology	-	16.3 {8.9}	
-geriatrics	11.7 {4.3}	-	14.7 {7.9}

Data are presented as the mean {SD} and frequency (%).

**Table 2 cancers-14-01386-t002:** Physicians’ care and perception of elderly cancer patients.

Care of Elderly Cancer Patients	Geriatricians (*n* = 7)	Oncology Physicians (*n* = 15)	Total (*n* = 22)
Elderly patients in active file	75{20.4}	31{19.5}	
Patients receiving:			
-oncology care	67%	-	
-geriatric care	-	38%	
Perception of the beginning of elderly age	75.2 (70–85) {6.1}	72 (60–80) {4.5}	73.5 (60–85) {5.1}

Data are presented as the mean {SD} or frequency % and range ().

**Table 3 cancers-14-01386-t003:** Processual determinants for oncogeriatric medical practice.

Processual Determinants	Geriatricians (*n* = 7)	Oncology Physicians (*n* = 15)
IDMs		
-weekly	-	-
-less than once a month	3 (50%)	9 (60%)
-bimonthly	1 (17%)	1 (7%)
-no interdisciplinary meetings	2 (33%)	5 (33%)
-no answer	1 (17%)	-
GA	7	5
-G8	2	3
-other types of GA	5	2

IDMs: interdisciplinary meetings; GA: geriatric assessment.

## Data Availability

The data presented in this study are available on request from the corresponding author. The data are not publicly available due to ethical reasons.

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
