# Peer review of "Exploring Determinants of Interdisciplinary Collaboration within a Geriatric Oncology Setting: A Mixed-Method Study"

_cancers, 2022, doi:10.3390/cancers14061386_

Round 1
Reviewer 1 Report
The authors used an original approach to explore the difficulties in implementing an inter professional collaboration for the care of geriatric patients in oncology.
My major concern is that they explored interspecialty and not inter professional collaboration. They do not really justify this choice and present in the introduction the interprofessional teamwork diagram, which is not what they really explore in their article. The only professionals involved in the study are physicians in the oncology and geriatrics departments. While this does not mean the study is uninteresting, it meeds to be acknowledged as a limitation and also to be reflected in the title.
Why did the authors consider that interviewing physicians was sufficient, when they reproduce the Reeves et al. diagram which clearly mentions professional power, hierarchy, professional roles, task shifting which suggests the need to look for answers beyond the medical profession.
Since this is a mixed methods study, the ARTS checklist should be used.
Author Response
- My major concern is that they explored interspecialty and not inter professional collaboration. They do not really justify this choice and present in the introduction the interprofessional teamwork diagram, which is not what they really explore in their article. The only professionals involved in the study are physicians in the oncology and geriatrics departments. While this does not mean the study is uninteresting, it needs to be acknowledged as a limitation and also to be reflected in the title.
Answer: We would like to thank the reviewer for taking the time to assess our manuscript and for raising this fundamental theoretical point. The word interprofessional has been replaced by “interdisciplinary” in the title and in the full manuscript. For clarity purposes, the term “teamwork” in the title has been replaced by “collaboration”. In the keywords section, the word “interprofessional teamwork” has been modified to “interdisciplinary collaboration”. In the line 103 the sentence “Considering the recommended oncogeriatric activities as interprofessional teamwork” has been deleted. Furthermore, we have revised the text to address your concerns, explaining the reasons that brought the authors to apply the framework of interprofessional teamwork of Reeves to the interdisciplinary collaboration between oncology physicians and geriatricians. Please see page 2 of the revised manuscript, lines 88–105.
We have already acknowledged this point in the limitations (page 9, lines 376-380), but we understand the need of explaining this point in the introduction.
- Why did the authors consider that interviewing physicians was sufficient, when they reproduce the Reeves et al. diagram whichclearly mentions professional power, hierarchy,professional roles, task shifting which suggests the need to look for answers beyond the medical profession.
Answer: This study is an ancillary study of Oncodage, a prospective multicenter cohort study, which had as objective the validation of G8 assessment tool. The aim of the current manuscript is to explore some of the contextual and processual determinants of the collaboration between geriatricians and oncology physicians. However, the aim of the study was not intended to be a full reproduction of the framework of Reeves. This is explained in the limitations section of the manuscript, page 9 lines 376-380: “This study carries several limitations. First of all, being an ancillary study of the ONCODAGE Prospective Multicenter Cohort Study, not all outcomes of interest could be explored. In fact, as suggested by the framework of Reeves et al. [34], other determinants should be assessed in order to fully comprehend the functioning of interprofessional teamwork.”.
However, thanks to the remark of the reviewer, we have noted a lack of clarity on this aspect and we have made some modifications in the manuscript. In the abstract, the sentence explaining this aspect has been modified to: “The multidimensional framework for interprofessional teamwork of Reeves has been used to analyse some of the determinants of the collaboration between oncology physicians and geriatricians”.
- Since this is a mixed methods study, the ARTS checklist should beused.
Answer: Following the suggestions of both reviewers we have decided to follow the Mixed Methods Article Reporting Standards (MMARS) of the American Psychological Association, JARS-Mixed. The expression “qualitative study” has been replaced in the keywords section to “mix-method study”, in order to facilitate understanding of the methodology used. Furthermore, we have reviewed the manuscript to respond to all of the 32 items of the COREQ checklist applicable to surveys.
Moreover, we have revised and extended the materials and methods introductory section and hope that it is now clearer. Please see this section on page 3 of the revised manuscript, lines 111-133.
Thank you again for the time that you have taken to review the manuscript and for the aspects to which you have pointed out.
Reviewer 2 Report
General
This paper addresses a vitally important issue. However, it is not well written. It is also severely limited by a small sample size and low response rate, which is not discussed. This appears to be a student paper, as there are some very basic issues with the language and format. There are many indicators of ESL grammar issues. Some sentences make no sense. The use of “if” is problematic. The paper needs a full English proof.
Specific
The Simple summary is too nonspecific, it should be more focused on this study.
Abstract
There are indicators of ESL grammar issues. Line 21 “If ”is an odd word to use here, maybe use “While” or “Although”.
Line 27-30: This sentence is too long, so is not easily comprehended. Write into two or more sentences with purpose.
Line 28: “particularly” is not needed, delete.
Introduction
Line 38: I do not understand what you mean by :“cancers’ limited medical knowledge”, Cancer per se has no knowledge. Its a disease.
”If ” is used throughout in many places that where it is not appropriate, e.g., line 77, line 89, line 224, line 345. ”If ” is a word that is not generally used this often. It seems to be an ESL mistranslation.
Figure 1 is not referred to in the text. Why is it there?
Methods
The sample size and response rate are small.
Line 115: delete “and their responses were included in data analysis.”
Line 127; what is “elderly hood”? This is extremely pejorative language.
The data analysis needs much more explication. The methods description should allow replication. Note that qualitative coding is usually agreed by researcher discussion and consensus not a quantitative formula. Please refer to the COREQ guidelines for the detail needed in describing qualitative work (https://www.equator-network.org/reporting-guidelines/coreq/)
Results
Line 183. Delete “A total of”. Delete “while” and insert “and”
Line 203 to 205 I do not understand what “in the active file” means. This should be one sentence.
Line 201. Find another expression for “elderly hood”
Line 220. Delete “Regarding factors impacting medical decisions”
Line 228. Delete “Regarding the management of older patients with cancer”
Line 245. “of” should be “for”
Line 246. “to not use” should be “not using”
Line 252- 265. Many language errors
Discussion
The small sample size is not noted as a limitation. This should be discussed.
Conclusions
This section should highlight new findings from this study and suggested directions for practice, policy, and research, not dictate what should be.
Author Response
We would like to thank the reviewer for taking the time to assess our manuscript. We have addressed all the concerns raised and the revised manuscript has been checked by two native English-speaking colleagues with full professional proficiency in English. We apologise for not having done it before sending it to Cancers.
Specific:
- The Simple summary is too nonspecific, it should be more focused on this study.
Answer: Following the suggestion of the reviewer, we have modified the simple summary making it more specific and focused on the current study. To do so, we have addressed the aspects recommended by the template provided by Cancers: 1) to explain why the research is being suggested, 2) what the authors aim to achieve, 3) and how the findings from this research may impact the research community. We hope that it is now clearer. Please see page 1 of the revised manuscript.
Abstract:
- There are indicators of ESL grammar issues. Line 21 “If ”is an odd word to use here, maybe use “While” or “Although”.
Answer: We thank the reviewer for noticing the ESL grammar issues contained in the manuscript. In the line 21, “If” has been replaced by “although”.
- Line 27-30: This sentence is too long, so is not easily comprehended. Write into two or more sentences with purpose.
Answer: The sentence of page 1 (lines 27-30) has been modified and separated into two sentences: “The multidimensional framework for interprofessional teamwork of Reeves has been used to analyze some of the determinants of the collaboration between oncology physicians and geriatricians. This study has identified systematic issues to consider when promoting communication and common values between the two disciplines, including available resources in terms of shared time, space and routine actions”.
- Line 28: “particularly” is not needed, delete.
Answer: “particularly” has been deleted.
Introduction:
- Line 38: I do not understand what you mean by :“cancers’ limited medical knowledge”, Cancer per se has no knowledge. Its a disease.
Answer: The sentence has been modified to: “to the limited medical knowledge of the disease at this age and to the need of adapted treatment in older patients” (page 1, lines 38-39).
- ”If ” is used throughout in many places that where it is not appropriate, e.g., line 77, line 89, line 224, line 345. ”If ” is a word that is not generally used this often. It seems to be an ESL mistranslation.
Answer: Thank you for pointing this out. We have modified all the sentences using the expression “if” with more appropriate words. The sentence “if it is feasible to incorporate geriatric-based competencies into the oncology setting” has been deleted.
- Figure 1 is not referred to in the text. Why is it there?
Answer: We have added the reference of the Figure 1 at line 101 of the manuscript.
Methods:
- The sample size and response rate are small.
Answer: We are aware of these limitations. However, considering the lack of articles published on this topic, we think that these results are still relevant for the community. As suggested by the reviewer, we have added a sentence in the limitation paragraph of the manuscript to discuss this aspect: “A further aspect to consider is the small sample size of the study and the low response rate that we have registered, which could have affected the results”.
- Line 115: delete “and their responses were included in data analysis.”
Answer: Following the suggestion of the reviewer, the sentence had been deleted.
- Line 127; what is “elderly hood”? This is extremely pejorative language.
Answer: We apologize for our error. The expression “elderly hood” has been replaced by “elderhood”.
- The data analysis needs much more explication. The methods description should allow replication. Note that qualitative coding is usually agreed by researcher discussion and consensus not a quantitative formula. Please refer to the COREQ guidelines for the detail needed in describing qualitative work (https://www.equator-network.org/reporting-guidelines/coreq/)
Answer: This study was a mix-method study, using qualitative and quantitative data. For this reason, we have decided to follow the Mixed Methods Article Reporting Standards (MMARS) of the American Psychological Association, JARS-Mixed. The expression “qualitative study” has been replaced in the keywords section to facilitate the understanding of the methodology used.
However, we have reviewed the article to make sure that it is in line with all the points of the 32 items COREQ checklist applicable to the methodology used in the study (online survey).
As suggested by of both reviewers, we have revised and extended the materials and methods introductory section and hope that it is now clearer. Please see this section on page 3 of the revised manuscript.
Results:
- Line 183. Delete “A total of”. Delete “while” and insert “and”.
Answer: The sentence has been modified following the suggestion or the reviewer: “Of the 22 participants, 7 geriatricians worked in the geriatric setting, and 15 physicians worked in the oncology setting (9 oncologists, 2 radiotherapists, 2 surgeons, 1 gastroenterologist, and 1 anesthetist)” (page 5, lines 210-214).
- Line 203 to 205 I do not understand what “in the active file” means. This should be one sentence.
Answer: We apologize for our error. The term “active file” has been removed as it was a mistranslation from French. The sentence has been modified: “Geriatricians on the other hand, reported greater oncogeriatric care for their older patients, with 67% of their patients receiving oncogeriatric care (Table 2)” (page 6, lines 230-232).
- Line 201. Find another expression for “elderly hood”.
Answer: “elderly hood” has been replaced by “elderhood”.
- Line 220. Delete “Regarding factors impacting medical decisions”
Answer: Following the suggestion of the reviewer the sentence has been deleted.
- Line 228. Delete “Regarding the management of older patients with cancer”
Answer: Following the suggestion of the reviewer the sentence has been deleted.
- Line 245. “of” should be “for”
Answer: “of” has been replaced by “for”.
- Line 246. “to not use” should be “not using”
Answer: The sentence has been modified following the suggestion of the Reviewer.
- Line 252- 265. Many language errors
Answer: We have revised the paragraph to correct language errors and improve the clarity (page7, lines 279-289): “All physicians reported making their medical decisions about elderly patients after consulting other specialties (e.g., direct interdisciplinary collaboration, interdisciplinary meetings). However, both geriatricians and oncology physicians reported a variability in regards to the frequency in which interdisciplinary meetings were organized. Overall, 33% of participants reported not having IDMs.
In the group of oncology physicians (n = 15), 7% stated having IDMs once a month and 60% had IDMs weekly. Moreover, 5 physicians out of 9 attends IDMs systematically, 2/9 frequently and 2/9 rarely.
In the group of geriatricians (n = 7), 17% systematically participated in bimonthly IDMs and 50% (n = 3) had weekly IDMs (to which 1 attended rarely, 1 frequently and 1 systematically)”.
Discussion:
- The small sample size is not noted as a limitation. This should be discussed.
Answer: We have added a sentence in the limitation paragraph of the manuscript (page 9, lines 381-383 to discuss this aspect “A further aspect to consider is the small sample size of the study and the low response rate that we have registered, which could have affected the results”.
Conclusions:
- This section should highlight new findings from this study and suggested directions for practice, policy, and research, not dictate what should be.
Answer: We have modified the conclusions following the suggestion of the Reviewer. The word “should” has been replaced by more appropriated expressions. Please see this section on page 9 of the revised manuscript, lines 390-401.
Round 2
Reviewer 2 Report
Thank you. Adequate changes have been made.